# From Faithfulness to Correctness: Generative Reward Models that Think Critically

## Abstract

Through reinforcement learning with verifiable rewards (RLVR), large language models have achieved substantial progress in domains with easily verifiable outcomes, such as mathematics and coding. However, when applied to more complex tasks like open-domain question answering, RLVR faces significant challenges due to the difficulty of verifying correctness. The nuanced and ambiguous nature of real-world knowledge makes it difficult to reliably evaluate correctness in these settings, necessitating further abilities that extend beyond mere logical consistency to encompass an understanding and assessment of both external and internal knowledge. Recent work has primarily focused on improving faithfulness, defined as semantic alignment with supporting documents, which can cause models to rely excessively on external sources and diminish their capacity for critical assessment. To address this, we propose the Thinking-supervised Reward Model (TRM), which incorporates sentence-level thinking supervision to endow reward models with *critical thinking* abilities. Given a query, answer, and supporting documents, TRM first assesses the faithfulness of each answer sentence to the supporting documents, and then applies a reasoning step to evaluate sentence-level correctness. By structuring reward modeling as a sequence of faithfulness, reasoning, and correctness evaluations, TRM encourages models to critically assess and leverage both external and internal knowledge. Experiments on reward signals demonstrate that TRM substantially improves the identification of incorrect sentences, and incorporating TRM into policy optimization leads to significant gains in both answer correctness and usefulness. We open-source our TRM model and implementation anonymously at https://anonymous.4open.science/r/TRM.

## 1 Introduction

Large language models (LLMs) have demonstrated impressive capabilities in step-by-step reasoning, especially when trained with reinforcement learning with verifiable rewards (RLVR) (Lambert et al., 2024; Guo et al., 2025). RLVR is highly effective in domains like mathematics and coding, where each reasoning step can be objectively verified and reward signals are unambiguous (Paul et al., 2023; Wu et al., 2023; Wang et al., 2023a; Ma et al., 2023; Khalifa et al., 2025; Zhang et al., 2025). However, this success does not easily extend to real-world tasks such as open-domain question answering, where verifying the correctness of each statement is challenging. This highlights a critical issue: language models struggle with complex reasoning unless logical connections are explicit and straightforward. Providing reward signals alone is far from sufficient for fostering genuine understanding of complex thinking patterns.

For example, consider the question: *"When was the novel 1984 written?"* with supporting document: *"Novel 1984 was published in 1949"* and answer: *"1984 was written by George Orwell in 1949"*. This answer is incorrect, as the novel *1984* was actually written in 1948 but published in 1949. Simply providing a negative reward may cause the model to incorrectly judge the supporting document as false, rather than recognizing the real error—overreliance on a correct but misleading external source. Verification is challenging because errors can stem from multiple, nuanced issues such as the relevance and accuracy of supporting documents, the model's faithfulness to those sources, and the avoidance of hallucination. While current RLVR techniques perform well in domains with clear rules, where errors primarily stem from logical reasoning, they are insufficient for complex, knowledge-intensive tasks that involve diverse and subtle types of errors.

Existing approaches to open-domain question answering often conflate the concepts of faithfulness and correctness. Faithfulness denotes semantic alignment between the generated answer and external supporting documents, whereas correctness concerns the factual accuracy of the answer. Although recent work has highlighted the importance of faithfulness (Durmus et al., 2020; Adlakha et al., 2024) and introduced methods to improve alignment with supporting documents (Zhao et al., 2024; Li et al., 2025), these solutions tend to overemphasize such documents without critically assessing their relevance or reliability. Consequently, models may rely excessively on supporting documents and fail to leverage their own internal knowledge, undermining overall answer quality.

To address this limitation, we propose the Thinking-supervised Reward Model (TRM), a sentence-level reward model designed to equip language models with *critical thinking* abilities to distinguish faithfulness from correctness and critically assess supporting documents. Given a query, answer, and supporting documents, TRM first evaluates the sentence-level faithfulness of the answer to the provided evidence. Building on the initial faithfulness assessment, TRM subsequently performs a reasoning step that explicitly examines how this assessment informs the factual correctness of each sentence, ultimately leading to a final correctness score. By structuring reward evaluation as a sequential process *(faithfulness → reasoning → correctness)*, TRM promotes a thinking pattern that first consults external sources to evaluate faithfulness, followed by the application of internal reasoning to determine the relationship between faithfulness and correctness. This methodology encourages reward models to leverage both external and internal knowledge, thereby enabling a clearer distinction between factual correctness and document faithfulness.

We conduct two stages of experiments. First, we evaluate TRM's ability to identify both sentence-level and answer-level errors on unseen data, demonstrating consistent improvements over outcome-supervised reward models (ORM) and process-supervised reward models (PRM). Second, we incorporate TRM into policy optimization within a reinforcement learning (RL) framework, where TRM ensures correctness and an auxiliary model addresses usefulness. Experiments on both the internal dataset and open-source out-of-distribution datasets show that our approach achieves substantial gains, improving correctness by up to 30.3% and increasing usefulness up to 35%.

Our main contributions are as follows:

- We propose a reward modeling framework that endows models with *critical thinking* ability by adopting sentence-level *faithfulness → reasoning → correctness* evaluation pattern.

- We offer an effective framework to reward modeling and policy optimization for open-domain question answering tasks where verification is challenging.

- To promote further research, we open-source our thinking-supervised reward model (TRM) and its implementation for policy optimization.

## 2 THINKING-SUPERVISED REWARD MODEL

### 2.1 SENTENCE-LEVEL REWARD SIGNALS

We address open-domain question answering (QA) with supporting documents. In many cases, an answer's overall quality is compromised not because every statement is incorrect, but due to the presence of specific sentences containing false or misleading information. Relying solely on outcome-supervised reward models (ORM) for training would inadvertently penalize such partially correct answers.

Prior work has shown that process-supervised reward models (PRM), which provide reward signals step by step, outperform outcome-supervised approaches in domains such as mathematics and coding (Lightman et al., 2023; Ma et al., 2023; Zhang et al., 2025). In these settings, correctness is objective and readily verifiable, making it relatively straightforward for both human annotators and LLMs to identify errors, which often take the form of explicit logical flaws or factual mistakes. Similarly, for question answering with supporting documents, we provide sentence-level supervision by assigning correctness labels to individual sentences, enabling more precise localization of relevant information. The process for constructing these supervision signals is detailed in Section 3.1.

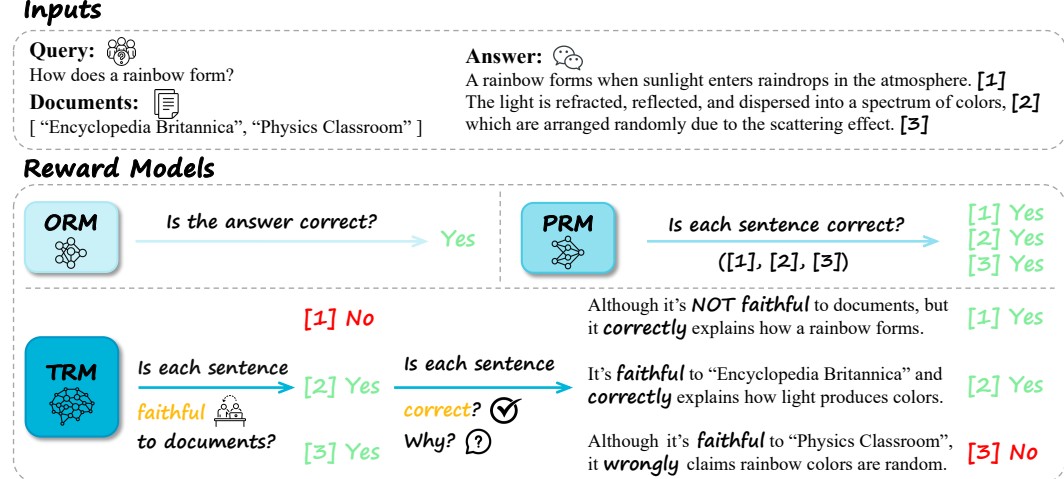

Figure 1: The overall architecture of outcome-supervised reward model (**ORM**), process-supervised reward model (**PRM**) and our thinking-supervised reward model (**TRM**). Given a query, documents, and an answer, TRM first assesses the faithfulness of each answer sentence to the documents, then evaluates its correctness through a reasoning step.

## 2.2 FAITHFULNESS → REASONING → CORRECTNESS

As our research progresses, we observe that providing models with only a final correctness signal is insufficient, especially for complex samples where simply updating internal knowledge does not guarantee generalization to unseen problems. This leads us to realize that our true objective is not to impart specific knowledge to reward models, but rather to teach LLMs effective thinking patterns. To achieve this, we manually design and implement a structured reasoning process that can be explicitly learned by language models. By guiding models through each step of this thinking pattern during training, we aim to foster more robust and transferable reasoning capabilities.

Inspired by the way humans evaluate answers, we observe that individuals typically begin by assessing whether an answer matches the supporting documents—a procedure that LLMs has managed to emulate. However, unlike LLMs, humans further engage in internal reflection, balancing trust in supporting documents with their own knowledge—a process often characterized as *critical thinking*. To formalize this distinction, we define "correctness" as the objective accuracy of an answer sentence, and "faithfulness" as the extent to which an answer aligns with the supporting documents. In practice, however, we find that models often conflate these two concepts. This conflation arises because models tend to assume that supporting documents are always correct (Huang et al., 2024; Wang et al., 2023b; Jin et al., 2024), thereby preventing them from distinguishing correctness from faithfulness, which is also the major problem of existing methods.

To address this, we design a structured thinking process that requires the model to explicitly reason from faithfulness to correctness, mirroring the critical thinking process humans use when evaluating information. Specifically, for each sentence in the answer, we first ask the model to assess whether the answer is faithful to the supporting documents—does it accurately reflect the evidence provided? Next, based on the faithfulness judgment, the model is further prompted to use its internal knowledge to assess whether the answer is factually correct through a reasoning step. Appendix C provides case studies demonstrating that this guided reasoning process leads to the development of more robust *critical thinking* abilities. Rather than relying solely on external sources, models begin to reflect internally when judging the correctness of answer sentences. As a result, models that internalize this evaluation logic achieve substantial improvements on complex question-answering tasks.

## 2.3 TRAINING STRATEGIES

Having established our framework (see Figure 1), we now describe the training strategies employed to instill the desired thinking patterns in language models. We adopt a two-stage process: initial su-

pervised fine-tuning (SFT) to teach the explicit reasoning steps, followed by reinforcement learning (RL) to further enhance the model's prediction abilities using both faithfulness and correctness as reward signals.

**Supervised Fine-tuning.** We begin by training the language model on our curated dataset, which is explicitly structured to mirror the thinking process *(Faithfulness → Reasoning → Correctness)* for each answer sentence. This sentence-level supervision not only enables the model to grasp the definitions of faithfulness and correctness, but also to understand how to logically progress from one to the other. For each query-answer pair, we present the model with the query, supporting documents, and the candidate answer, where the answer is segmented into sentences using predefined rules (see Appendix B.1 for details). These form the input to the model. *Faithfulness → Reasoning → Correctness* is used as target directly for SFT.

During training, we mask the input and optimize the standard negative log-likelihood (NLL) loss:

$$\mathcal{L} = -\frac{1}{N} \sum_{i=1}^{N} \sum_{t=1}^{T_i} \log p_\theta(y_t^{(i)} \mid y_{<t}^{(i)}, x^{(i)}) \tag{1}$$

where $x^{(i)}$ denotes the input for the $i$-th query-answer pair, $y_t^{(i)}$ is the target token at position $t$, and $y_{<t}^{(i)}$ represents the preceding target tokens.

By exposing the model to examples that decompose the reasoning process into interpretable steps, SFT facilitates subsequent stages of training, as it guides the model to produce not only correct answers, but also develop an interpretable and reliable reasoning trajectory.

**Reinforcement Learning.** To further empower the model to explore novel reasoning paths beyond those demonstrated during SFT, we introduce a reinforcement learning (RL) phase as a subsequent stage of training. Conventional RL approaches for language models typically rely solely on the final correctness of an answer as the reward signal. However, our analysis of post-SFT performance highlights a key observation: when a model learns to accurately assess the faithfulness of each intermediate reasoning step, its probability of producing correct answers increases significantly (see results in Section 3.3). Building on this insight, we integrate faithfulness as an intermediate reward in addition to the correctness signal within our RL framework, with both rewards assessed at the sentence level. This dual-signal strategy encourages the model not only to arrive at correct answers, but also to do so via faithful and interpretable reasoning trajectories. In turn, this fosters both greater robustness and enhanced accuracy in label prediction.

Formally, for sentence $k$ in query-answer pair $i$, given the faithfulness score $f(i, k)$ and correctness score $c(i, k)$, we define the reward function as

$$r_{i,k} = c(i, k) + \alpha \cdot f(i, k), \tag{2}$$

where $\alpha$ is a hyperparameter that balances the two reward components. In our experiments, we set $\alpha = 0.5$. Additionally, due to the strong bias toward correct labels in our dataset, we provide an extra reward when both the prediction and the ground truth label are incorrect (vice versa). Detailed analysis is provided in Section 3.4. This reward is then used within the Group Relative Policy Optimization (GRPO) algorithm (Guo et al., 2025) to optimize the policy model.

## 3 REWARD MODEL EXPERIMENTS

### 3.1 DATA CONSTRUCTION

We collect anonymized queries from Tencent WeChat search engine, which encompasses a comprehensive range of topics related to everyday knowledge. To safeguard user privacy, all data undergoes rigorous anonymization processes, removing any personally identifiable information prior to analysis. Before annotation, we employ rule-based methods to segment answers into individual sentences for finer granularity (details available in Appendix B.1). Our dataset is originally in Chinese, with all case studies and samples in this paper automatically translated into English for clarity.

Our data annotation process consists of two stages. In the first stage, human annotators assess whether each sentence is faithful to the supporting documents, labeling each sentence as either

Table 1: Reward model performance comparison. The best results are in bold.

|          | ORM    | PRM    | TRM-   | TRM    | TRM+   | Δ      |
|----------|--------|--------|--------|--------|--------|--------|
| F1 Score | 0.2564 | 0.3194 | 0.3238 | 0.3384 | **0.3447** | +6.5%  |
| Detection| 0.3222 | 0.3479 | 0.3483 | 0.3643 | **0.3690** | +5.9%  |

faithful or unfaithful based on its alignment with the source material. In the second stage, a separate annotator reviews the query, supporting documents, answer, and the initial faithfulness label. This annotator is responsible for verifying the factual correctness of each sentence, using but not solely relying on the faithfulness labels. To ensure well-informed judgments, annotators are encouraged to consult a variety of resources, including web searches, authoritative references, and domain-specific databases. Additionally, annotators provide a brief rationale for their correctness decision, beginning from considerations of faithfulness, which serves as the reasoning chain for the annotation.

This dual-stage annotation framework enables us to distinguish between several nuanced scenarios:

- *Faithful and Correct:* The answer aligns with supporting documents and is factually correct.
- *Unfaithful but Correct:* The answer doesn't align with supporting documents but remains factually correct. This can occur when supporting documents contain misleading information.
- *Faithful but Incorrect:* The answer aligns with supporting documents, but the content itself is incorrect, possibly due inaccuracies within the source materials or outdated information.
- *Unfaithful and Incorrect:* The answer neither aligns with supporting documents nor is factually correct, potentially arising from hallucinations, fabrication, or misunderstanding.

By disentangling faithfulness and factual correctness through this annotation process, we create a rich and robust dataset. This enables the study of complex real-world scenarios where retrieved evidence may be unreliable, incomplete, or contradictory, and supports the development of models that can reason about both the provenance and truthfulness of information. Details of the dataset can be found in Appendix A.1.

### 3.2 EXPERIMENTAL SETUP

**Evaluation.** It is important to note that the majority (86.86%) of answers in our dataset are labeled as correct, consequently, our primary focus is on evaluating the model's ability to distinguish incorrect answers from correct ones. This is particularly relevant since a single query may have multiple fully correct answers (i.e., answers with no incorrect sentences). Our evaluation therefore emphasizes the detection of incorrect content and adopts two main metrics:

- *F1 Score (Incorrect Sentences).* We compute the F1 score specifically for sentences labeled as incorrect. Since correct sentences are much more prevalent, the overall F1 score is heavily biased toward the correct class, making metrics such as F1 (overall) or F1 (correct) less informative—indeed, always predicting "correct" would yield artificially high scores.
- *Detection (Incorrect Answer).* Given that we have both ground-truth and predicted correctness labels for each sentence within an answer, we compute the proportion of incorrect sentences in each answer and use this as the answer's correctness score. We then assess whether our reward model can accurately detect the worst answer (i.e., the answer with the lowest correctness score) among all candidate answers for the same query.

More metrics including F1 (overall), F1 (correct), Recall and NDCG are reported in Appendix A.2.

**Baselines.** We compare our thinking-supervised reward model (TRM) with two baselines: the outcome-supervised reward model (ORM) (Uesato et al., 2022) and the process-supervised reward model (PRM) (Lightman et al., 2023). The ORM is trained based on answer-level correctness scores, following the same procedure outlined previously in the evaluation. During evaluation, the predicted answer-level score is uniformly assigned to each sentence when computing the sentence-level F1 score. The PRM is trained with sentence-level correctness scores, employing a generative approach to produce answer labels.

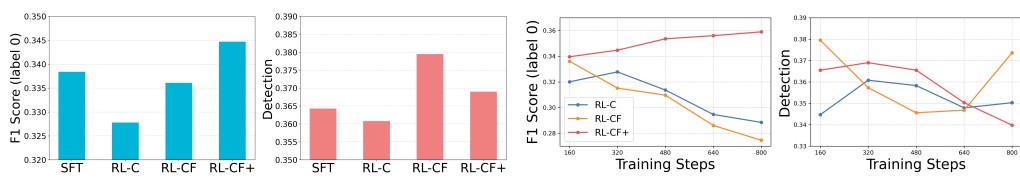

(a) Ablation study on model variants.    (b) Reward model metrics across training steps.

Figure 2: Analysis towards reinforcement learning.

Additionally, we include three variants of TRM for both fair comparison and ablation analysis. Since both ORM and PRM are trained solely with SFT, we denote the SFT-only variant of our model as TRM, and the variant further trained with reinforcement learning as TRM+. To investigate the contributions of different supervision signals, we introduce a variant without the reasoning path, denoted as TRM-, which serves as an ablation to assess the impact of the reasoning component.

### 3.3 MAIN RESULTS

As shown in Table 1, our proposed thinking-supervised reward model (TRM) consistently outperforms all baselines across all evaluation metrics. These results demonstrate the effectiveness of introducing thinking-supervised patterns into the reward model design. The improvements reported in the $\Delta$ column reflect the relative gain of TRM over the best non-TRM variant. Statistically, it is challenging to detect incorrect sentences or answers, as they tend to occur infrequently compared to correct ones.

In our ablation study, the TRM variant without explicit reasoning paths (TRM-) underperforms compared to the full TRM, confirming the importance of incorporating explicit reasoning steps into the model's architecture. Notably, TRM- still surpasses PRM, providing direct evidence that the addition of faithfulness and mid-point thinking patterns contributes to improved judgment structure. Overall, these findings substantiate the contribution of each individual component in our thinking-supervised pattern design.

### 3.4 ANALYSIS

It is important to note that during continued reinforcement learning (RL) training, we employ both faithfulness and correctness as reward signals, along with an additional reward assigned for successfully predicting incorrect labels (as described in Section 2.3). To further investigate the contribution of each of these reward signals during RL, we evaluate the following four model variants:

- SFT: Model trained with supervised fine-tuning only, without reinforcement learning.

- RL-C: Model trained with supervised fine-tuning followed by reinforcement learning, using correctness as the sole reward signal.

- RL-CF: Model trained with reinforcement learning where the reward signal combines both correctness and faithfulness (weighted 2:1).

- RL-CF+: RL-CF with an additional reward for correct prediction of incorrect labels and an extra penalty for failure to predict incorrect labels.

The results in Table 2a summarize the performance of each model variant in terms of F1 score for sentence-level incorrect label and answer-level incorrect answer detection rate. Table 2b further illustrates the evolution of these two metrics on test datasets throughout the course of reinforcement learning (RL) training. Notably, only the RL-CF+ variant consistently outperforms the SFT baseline, while RL-CF demonstrates superiority solely in answer-level detection rate, and RL-C underperforms across both metrics. We also observe that as training progresses, there is a consistent increase in sentence-level F1 score (label 0) exclusively for RL-CF+, whereas the other two variants, which lack an additional reward for incorrect labels, exhibit a consistent decrease. In contrast, the answer-level detection rate remains relatively stable across all variants. Below, we provide a detailed analysis from two perspectives: (1) Why does RL sometimes hurt performance (RL-C and RL-CF

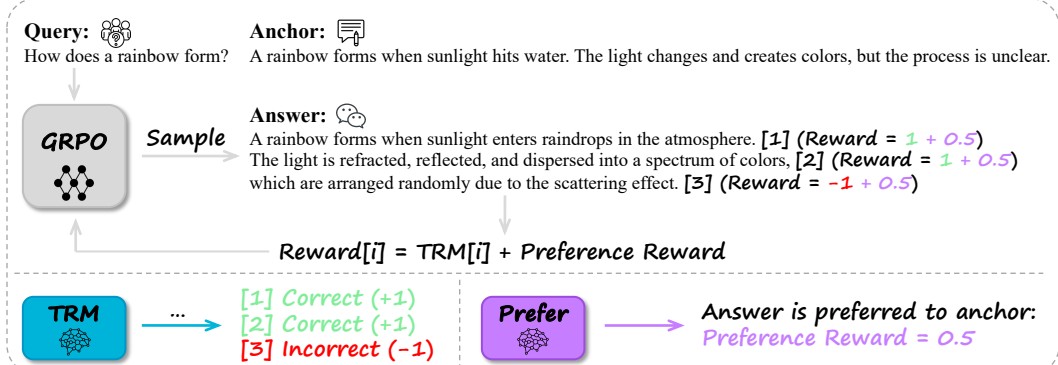

Figure 3: The overall architecture of policy optimization with the thinking-supervised reward model (**TRM**) and the preference reward model (**Prefer**). TRM provides sentence-level reward signals for correctness, while the preference reward model supplies an answer-level preference score that is broadcast to each sentence. For each sentence, these rewards are combined and then used as input to the GRPO algorithm to update the policy model.

occasionally worse than SFT)? (2) Why does introducing an additional reward for incorrect labels improve performance?

First, RL can underperform because it often overfits to training data by optimizing narrow reward signals, reducing generalization. In our sentence-level reasoning task, the path from faithfulness to correctness is mostly straightforward, with little room for exploration or alternative solutions. Unlike domains such as mathematics or coding, where multiple plausible solution paths exist and exploration is crucial, our task offers little room for RL to discover new reasoning strategies. The reasoning path may not be divergent enough, and the incorrect point is generally well-defined, leaving few alternative or wrong reasoning trajectories. As a result, attempts to force RL to explore in this constrained space can lead to spurious reasoning patterns that coincidentally produce correct answers.

Second, incorporating an additional reward for identifying incorrect labels helps to mitigate the data imbalance problem, as a large majority of sentences (86.86%) are correct labels. Without this adjustment, the reward model may overfit to the dominant class. By providing extra reward when the model correctly identifies a mistake (and penalizing it otherwise), we encourage the model to focus more on potential errors, promoting a more balanced and attentive learning process. This adjustment not only prevents overfitting to the majority class but also encourages the model to reason more carefully about where mistakes may occur, which can lead to improved detection performance.

## 4 POLICY OPTIMIZATION

Having demonstrated that our thinking-supervised reward model (TRM) can reliably distinguish between correct and incorrect sentences, we now proceed to evaluate its effectiveness as a reward signal for policy model training within a reinforcement learning (RL) framework. An overview is provided in Table 3.

### 4.1 REINFORCEMENT LEARNING FRAMEWORK

In our reinforcement learning framework, we utilize the reward model to provide feedback signals for each rollout answer generated by the policy model. Specifically, TRM is responsible for evaluating the correctness of individual answers. While it is intuitive to use TRM directly as the generative reward model, it is important to note that TRM focuses solely on correctness, which represents only one aspect of overall answer quality.

For instance, a concise answer may always be correct, yet it might lack sufficient information to be considered useful. We refer to this broader dimension as *usefulness*. To address this, we adopt

a common practice of combining multiple reward signals: in addition to TRM, we incorporate a preference reward model (Prefer) that captures aspects beyond correctness.

For each query, we select an answer with a perfect TRM score as the "anchor". The preference reward model is then used to compare the rollout answer with the anchor, providing a positive reward if the rollout is preferred, and a negative reward otherwise. The detailed prompting strategy for the preference model is described in Appendix D.

The final reward signal for sentence $k$ in answer $i$ is computed as:

$$r_{i,k} = \text{TRM}(i, k) + \beta \cdot \text{Prefer}(g, i) \tag{3}$$

where $\text{TRM}(i, k)$ denotes the correctness score assigned by TRM, and $\text{Prefer}(g, i)$ is the preference score comparing answer $i$ to the golden answer $g$. Hyperparameter $\beta$ balances the contribution of usefulness and correctness. The combined reward signal is subsequently used within the Group Relative Policy Optimization (GRPO) algorithm (Guo et al., 2025) to optimize the policy model.

## 4.2 EXPERIMENTAL SETUP

**Datasets.** To rigorously evaluate our reward model, we ensure the policy model is tested on queries not seen during reward model training, thus preventing data leakage. Specifically, we curate a new dataset consisting of 9423 queries collected again from Tencent WeChat search engine. For evaluation, to ensure thorough assessment, we construct a challenging test set of 94 queries, each of which contains at least one incorrect answer to assess the robustness of the trained policy.

Although our commercial search engine offers a wide variety of question types and is regarded as a comprehensive, all-domain resource, we further evaluate the generalization capability of our reward model beyond in-domain data. To this end, we additionally assess our model on the open-domain CRUD dataset (Lyu et al., 2025), a Chinese-language dataset comprising queries, supporting documents, and reference answers. Specifically, we utilize 8618 query-answer pairs for training and reserve 100 pairs for testing.

**Evaluation.** As discussed in Subsection 4.1, we evaluate both *correctness* and *usefulness* of the generated answers. For correctness, we employ GPT-4.1 (OpenAI, 2024) to assess the correctness of each sentence within every answer. Specifically, we calculate the proportion of correct sentences as the answer's score and report the average across all test samples as the final correctness metric. We also report the number of completely correct answers, defined as answers in which all sentences are judged to be correct. For our constructed challenging dataset, we manually provide GPT-4.1 with several incorrect answers for each query and explicitly indicate where the mistakes occur, thereby enabling a more accurate evaluation. In contrast, for the CRUD dataset—which features an easier test set—GPT-4.1 alone is sufficient to identify the errors without additional manual intervention.

Evaluating usefulness is more nuanced, as it is a less well-defined and inherently more relative concept. To address this, we provide an anchor answer (sampled from Qwen2.5-32B-Instruct, also referred to as "Base") alongside the answer generated after training. We then employ GPT-4.1 to compare each answer to the anchor. To eliminate any bias arising from the order in which answers are presented to GPT-4.1—since LLMs may sometimes exhibit a preference for the first answer in a pair—we present both the anchor and the generated answer in both possible positions, conducting the evaluation twice for each pair. If GPT-4.1 consistently prefers one answer, regardless of its position, then it's counted as a win; otherwise, it is considered a loss. If no consensus can be reached, the comparison is marked as a tie. To ensure fairness across different models, the anchor answer remains fixed. Detailed evaluation prompts are provided in Appendix D.

## 4.3 MAIN RESULTS

Table 2 demonstrates that the joint use of both the preference reward model (Prefer) and the thinking-supervised reward model (TRM) leads to significant improvements in both correctness and usefulness. This can be attributed to the complementary strengths of the two reward models: preference reward model captures usefulness, while TRM focuses on correctness. Notably, for the Tencent test dataset, which deliberately includes challenging queries only, the joint model shows a substantial improvement in usefulness (from 33 to 43). In contrast, for the CRUD dataset, where test samples

Table 2: Evaluation results on the Tencent and CRUD datasets. The best results are in bold.

| Data | Tencent | | | | | CRUD | | | | |
|---|---|---|---|---|---|---|---|---|---|---|
| Metrics | Correctness | | Usefulness | | | Correctness | | Usefulness | | |
| | Count | Score | Win | Lose | Tie | Count | Score | Win | Lose | Tie |
| Base | 48 | 0.8876 | – | – | – | 66 | 0.9075 | – | – | – |
| Prefer | 53 | 0.8672 | 34 | 28 | 32 | 76 | 0.9182 | 76 | 6 | 18 |
| TRM | 50 | 0.8878 | 33 | 27 | 34 | 84 | 0.8945 | 79 | 7 | 14 |
| Joint | **59** | **0.9158** | **43** | **20** | 31 | **86** | **0.9489** | **85** | **3** | 12 |

are chosen randomly, the joint model achieves a significant improvement in correctness (from 66 to 86). These findings demonstrate the joint model's capacity to adapt to dataset complexity, excelling in usefulness for more difficult queries and in correctness for easier ones.

## 5 RELATED WORK

### 5.1 REINFORCEMENT LEARNING FROM VERIFIABLE REWARDS (RLVR)

Reinforcement learning from verifiable rewards (RLVR) is a key technique for encouraging large language models to perform step-by-step reasoning (Lambert et al., 2024; Guo et al., 2025). Recent advances in reward modeling have focused on three main directions. First, process-supervised reward models enhance feedback by assessing intermediate reasoning steps rather than just final answers (Uesato et al., 2022; Lightman et al., 2023). Second, fine-grained reward functions better capture nuanced human preferences, allowing models to align more closely with desired behaviors (Wu et al., 2023; Song et al., 2025). Third, explicit reasoning supervision helps models produce more coherent and robust chains of thought (Chen et al., 2025; Khalifa et al., 2025; Zhang et al., 2025). These approaches are particularly effective in domains with clear logic and verifiable outcomes, such as mathematics and coding, but they struggle to extend to more complex, open-ended domains, where our method demonstrates greater applicability.

### 5.2 GENERATIVE REASONING REWARD MODELS

Parallel to advances in reward modeling, researchers have explored leveraging chain-of-thought (CoT) reasoning (Wei et al., 2022) to enhance reward model predictions. However, as CoTs do not always faithfully reflect a model's underlying reasoning process (Chen et al., 2025; Turpin et al., 2023), a variety of approaches have emerged to address this limitation. These include incorporating intermediate reasoning steps (Paul et al., 2024), formulating reward modeling as a next-token prediction task (Zhang et al., 2024), ranking CoT traces (Wu et al., 2025b), and employing external verification tools (Lyu et al., 2023). In contrast, our approach provides explicit sentence-level supervision and feedback at each reasoning step, ensuring that the model's reasoning remains faithful to its true thought process, thereby increasing trust in its outputs.

## 6 CONCLUSION

We have introduced the thinking-supervised reward model (TRM), a novel framework for equipping language models with critical thinking abilities through structured, sentence-level reward evaluation. By modeling the reward process as a sequence of faithfulness, reasoning, and correctness, TRM effectively addresses the challenges of complex, open-domain tasks where verification is difficult and sources of error are nuanced. Our experiments demonstrate that TRM not only improves error detection at both the sentence and answer levels, but also enhances policy optimization in reinforcement learning settings, leading to significant gains in correctness and usefulness. We hope our open-sourced TRM and its implementation will inspire further research into more rigorous and interpretable reward modeling for language models.

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

# A DATA AND PERFORMANCE ANALYSIS

## A.1 REWARD MODEL DATASET

We detail the dataset used for the reward model, which is sourced from Tencent WeChat search engine. Table 3. includes the number of queries, answers, sentence counts as well as the proportion of correct and incorrect sentences (reported as percentages).

Table 3: Tencent dataset for reward model

|          | Query | Answer | Sentences | | |
|----------|-------|--------|-------|-------------|-------------|
|          |       |        | count | positive (%) | negative (%) |
| Overall  | 2133  | 8648   | 93322 | 86.86%      | 13.14%      |
| Training | 1919  | 7784   | 84068 | 86.75%      | 13.25%      |
| Testing  | 214   | 864    | 9254  | 87.86%      | 12.14%      |

## A.2 PERFORMANCE

As discussed in Section 3.2, we report the F1 score for incorrect sentences and the detection of incorrect answers. It is important to highlight that our dataset is highly imbalanced, with a significantly larger proportion of correct sentences. Accordingly, our primary objective is to demonstrate the ability to effectively improve the labeling of incorrect sentences while maintaining competitive performance on correct ones. To this end, we evaluate the metrics summarized below in Table 4, considering both correct and incorrect answers/sentences:

- *F1 Score (All Sentences).* We report the F1 score across all sentences, regardless of correctness.
- *F1 Score (Correct Sentences).* We report the F1 score for correct sentences only.
- *Recall.* We report the recall score across all sentences to demonstrate that our model does not result in false detection issues while excelling at identifying incorrect labels.
- *Normalized Discounted Cumulative Gain (NDCG).* We report the NDCG@4 score for each query (with 4 answers per query in the test set) based on ranked correctness scores. Since most answers are fully correct, NDCG remains consistently high with minimal variation, as top-ranked answers often receive perfect scores. Thus, even small NDCG differences can reflect significant improvements due to the limited answers and prevalence of perfect scores.

Table 4: Additional reward model performance comparison. The best results are in bold.

|              | ORM    | PRM    | TRM-       | TRM    | TRM+       |
|--------------|--------|--------|------------|--------|------------|
| F1 (overall) | 0.7274 | 0.8680 | **0.8715** | 0.8652 | 0.8648     |
| F1 (correct) | 0.8331 | 0.9268 | **0.9289** | 0.9240 | 0.9237     |
| Recall       | 0.7743 | 0.8677 | **0.8713** | 0.8635 | 0.8630     |
| NDCG         | 0.9036 | 0.9052 | 0.9047     | 0.9086 | **0.9091** |

Notably, TRM variants achieve better overall performance than ORM and PRM. However, accurately predicting correct labels is relatively easy, since consistently predicting "correct" naturally leads to high scores on these metrics. The primary focus here is to demonstrate our method enjoys a low false detection rate and does not significantly compromise its ability to predict correct labels.

# B IMPLEMENTATION DETAILS

We use Qwen2.5-32B-Instruct (Team, 2024) as the backbone architecture for our policy model. For all reward models, we employ DeepSeek-R1-Distill-Qwen-32B (Guo et al., 2025) as the underlying model. We select the checkpoint that achieves the highest detection rate of incorrect answers. Both supervised fine-tuning (SFT) and reinforcement learning (RL) are performed using the Tencent Wechat-YATT training framework (Wu et al., 2025a).

### B.1 THINKING-SUPERVISED REWARD MODEL

**Sentence separation.** The sentence separation process first divides each answer into segments using regular expressions that identify common sentence-ending punctuation marks. The script then further processes these segments to recognize and group structured elements such as headings, numbered lists, and bullet points, ensuring that these are treated as cohesive units rather than being split incorrectly. For every identified sentence or section, the original text is preserved and paired with an order label. In the final step, the script inserts explicit sentence order markers (e.g., [Sentence 1]) into the answer text, producing a version where each sentence or section is clearly delineated and labeled. The processed results are then saved for downstream analysis or annotation.

**Supervised Fine-tuning.** We perform supervised fine-tuning (SFT) for our TRM using the dataset described in Appendix A. Training is conducted with a batch size of 32 for 4 epochs on 128 NVIDIA H20 GPUs. We employ tensor parallelism, pipeline parallelism, and data parallelism with degrees of 8, 4, and 4, respectively. Each epoch requires approximately one hour of training. The Adam optimizer (Kingma & Ba, 2014) is utilized, with an initial learning rate of $3 \times 10^{-6}$, a minimum learning rate of $1 \times 10^{-7}$, and 200 warm-up steps.

**Reinforcement Learning.** We perform reinforcement learning (RL) on our TRM following supervised fine-tuning. RL is conducted with a batch size of 128 for 800 steps on 128 NVIDIA H20 GPUs, with each training session requiring approximately 24 hours. We employ tensor parallelism, pipeline parallelism, and data parallelism with degrees of 8, 4, and 4, respectively. The Adam optimizer is used, with an initial learning rate of $5 \times 10^{-8}$, a minimum learning rate of $5 \times 10^{-9}$, 20 warm-up steps, and a KL-loss coefficient ($\beta$) of 0.01. For sampling, we use a rollout batch size of 32 and apply top-$k$ sampling ($k = 40$) for decoding, with the temperature parameter set to 1.0.

### B.2 POLICY OPTIMIZATION

We perform reinforcement learning (RL) on the policy model using both the TRM and the preference reward model, with a reward weighting ratio of 1:2. RL is conducted with a batch size of 128 for 2240 steps on 128 NVIDIA H20 GPUs. Each training session requires approximately 72 hours for Tencent dataset and 24 hours for CRUD dataset. We employ tensor parallelism, pipeline parallelism, and data parallelism with degrees of 8, 4, and 4, respectively. The Adam optimizer is used, with an initial learning rate of $1 \times 10^{-7}$, a minimum learning rate of $5 \times 10^{-9}$, 20 warm-up steps, and a KL-loss coefficient ($\beta$) of 0.01. For sampling, we use a rollout batch size of 32 and apply top-$k$ sampling ($k = 40$) for decoding, with the temperature parameter set to 1.0.

## C CASE STUDIES

To highlight the unique strengths of our TRM in fostering critical thinking within LLMs, we present representative case studies for each of the following scenarios: (i) faithful and correct, (ii) faithful but incorrect, (iii) unfaithful but correct, and (iv) unfaithful and incorrect. Since including all document answers and sentence-level reasoning would be prohibitively lengthy, we selectively present only the most illustrative segments for each example. For clarity, the original texts (in Chinese) have been translated into English.

### C.1 FAITHFUL AND CORRECT

*QUERY:* What is the highest alcohol content of vodka?

*DOCUMENTS:* The highest alcohol content of vodka is 96%, found in rectified vodka from Poland (Spirytus Rektyfikowany). Most vodkas are typically 40-60% alcohol. Vodka is a distilled spirit made from fermented grains, known for its clean and pure taste ...

*ANSWER:* Vodka can reach a maximum alcohol content of 96%, most notably in the Polish Spirytus Rektyfikowany ...

*GROUND TRUTH:* The sentence is faithful to the source information and accurately describes the highest alcohol content of vodka and the specific product.

*TRM:*
{

    *"Faithfulness Score":* 1,
    *"Reason for Correctness Score":* "The sentence is faithful to the source, factually accurate and complete, describing the highest strength, representative brand, characteristics, and uses of vodka, without any errors or omissions.",
    *"Correctness Score":* 1

}

---

## C.2 FAITHFUL BUT INCORRECT

*QUERY:* How should the penalty for breach be calculated when the seller refuses to deliver goods?

*DOCUMENTS:* ... In cases where the seller fails to deliver goods, according to the provisions of the E-Commerce Law, the seller should compensate the consumer an amount ranging from three to ten times the payment made by the consumer ...

*ANSWER:* ... In cases of non-delivery, the E-Commerce Law purportedly requires compensation of three to ten times the purchase amount ...

*GROUND TRUTH:* The compensation standard should be based on the contract terms or actual losses, not three to ten times the amount. The sentence content contradicts the legal regulations.

*TRM:*
{

    *"Faithfulness Score":* 1,
    *"Reason for Correctness Score":* "The sentence cites the 'E-Commerce Law' and mentions a specific compensation ratio, but the 'E-Commerce Law' does not explicitly stipulate a specific ratio, so there is a factual error.",
    *"Correctness Score":* 0

}

---

## C.3 UNFAITHFUL BUT CORRECT

*QUERY:* How many calories are in 100g of ham sausage?

*ANSWER:* 100 grams of ham sausage contains about 212 kcal. Note that the calorie content may vary by brand, and as a processed meat product, its calories depend on ingredients and production methods ...

*GROUND TRUTH:* The information is correct. Although there is no direct source, the answer is reasonable, consistent with the fact that ham sausage is relatively high in calories, and includes a reminder to consume in moderation.

*TRM:*
{

    *"Faithfulness Score":* 0,
    *"Reason for Correctness Score":* "There is no direct source in the original text, but the content is correct and aligns with common knowledge.",
    *"Correctness Score":* 1

}

---

## C.4 UNFAITHFUL AND INCORRECT

*QUERY:* What is the purpose of teaching how to sketch a cube?

***ANSWER:*** ... By using the structural features of a cube, students can master techniques for combining square and round forms, providing a foundational skill for drawing more complex subjects like figures and scenes ...

***GROUND TRUTH:*** The answer contains factual errors, a cube does not help master the combination of square and round forms, and the information is not faithful to the original text.

***TRM:***
{

*"Faithfulness Score":* 0,
*"Reason for Correctness Score":* "The answer is not faithful to the source, and its claims about building skills for complex forms are incorrect.",
*"Correctness Score":* 0

}

## D   PROMPTS

In this section, we detail all prompts utilized throughout our experiments, including those for training reward models, guiding policy models, and evaluating responses via LLM-as-a-Judge. For clarity, the original texts (in Chinese) have been translated into English.

### D.1   PROMPTS FOR THINKING-SUPERVISED REWARD MODEL

You will receive information in the following format:

```
{
    "query": // user query
    "now_time": // time of query
    "search_result": // retrieved original text
    "answer_order": // the answer is split into sentences:
        "
            Sentence content 0 [Sentence 0]
            Sentence content 1 [Sentence 1]
            ...
        "
}
```

Please follow these steps:

1. For each sentence in `answer_order` (marked as [Sentence i]), perform:

   (a) Judge whether the sentence is faithful to the source text:
      - Faithful: The content aligns with the source, or is a reasonable summary, inference, or suggestion. Sentences without substantive information are also considered faithful.
      - Unfaithful: The content does not match the source or includes information not mentioned and cannot be reasonably inferred.

   (b) Judge the correctness, classified as:
      - Faithful and correct: Professional terms, data, specific conclusions, or cited opinions supported by the source.
      - Faithful but incorrect: The source itself is wrong, outdated, contains conflicting information, or contains common-sense errors.
      - Unfaithful but correct: Reasonable summary, general knowledge, or reasonable inference based on the source and common sense.
      - Unfaithful and incorrect: Fabricated information, excessive speculation, contradiction, etc. Only mark as incorrect if it cannot be reasonably inferred from context or common sense.

2. Output the results in the following `List` format:

```
[
    // For each sentence, use this format
    {
        "Faithfulness Score": 0 or 1, // 0: unfaithful, 1: faithful
        "Correctness Reason": "",
        "Correctness Score": 0 or 1 // 0: incorrect, 1: correct
    }
]
```

**Notes:**

- The order of sentences in the output `List` must exactly match the order in `answer_order`.
- Output only the final `List`, nothing else.

---

### D.2 PROMPTS FOR POLICY MODEL

You will receive information in the following format:

```
{
    "query": // user question
    "search_result": // retrieved original text
}
```

Please answer the "query" based on the "search_result". Follow these requirements:

- Understand Intent: Before answering, thoroughly analyze and understand the core intent of the user's question to ensure your answer directly addresses the user's needs.
- Complete Information: The answer should cover all key information, ensuring thorough and comprehensive content with no omissions.
- Clear Expression: Use concise and fluent language, avoid repetition and redundancy, and ensure the answer is easy to understand.
- Logical Structure: Present the answer in a clear and organized manner. Use bullet points, lists, or paragraphs as appropriate to structure the content logically.
- Reference the Original Text: Base your answer primarily on the "search_result"; when necessary, make reasonable inferences using your own knowledge.

Please process the following content according to the above rules:

---

### D.3 PROMPTS FOR CORRECTNESS EVALUATION

You are a professional and meticulous QA annotator. Your task is to analyze factual errors in the "Answer to be Evaluated" based on the "Question", "References", and your own model knowledge.

**Input Information:**

```
{
    "Question": The question or task to be answered.
    "References": Reference materials related to the question.
    "Answer to be Analyzed": The answer containing errors.
}
```

**Analysis Requirements:**

1. Analyze the factual errors in each sentence of the "Answer to be Analyzed", sentence by sentence.
2. Only analyze factual errors; do not consider issues of expression, structure, logic, or missing content.
3. Carefully check each sentence, and point out even the slightest factual errors. If there is a suspected factual error, mark and explain the uncertainty—do not ignore any possible factual errors due to uncertainty.

4. After the analysis, calculate the error ratio (number of erroneous sentences / total number of sentences) and output it in float format.

Output the result in the following `JSON` format (strictly follow the requirements, do not output anything extra):

```
{
    "Error Analysis": [
        // Error analysis for each sentence
    ],
    "Error Ratio": // float format
}
```

Please process the following content according to the above rules:

---

### D.4 PROMPTS FOR USEFULNESS EVALUATION

You are acting as a senior QA evaluator. Your task is to determine the preference order ("partial order") between "Answer 1" and "Answer 2" based on the "Question" and your internal model knowledge.

**Input Information:**

```
{
    "Question": // The question or task to be answered
    "Answer 1": // The first answer
    "Answer 2": // The second answer
}
```

**[Usefulness Definition]**

• 0 points – Poor: The answer provides information irrelevant to the question's requirements.

• 1 point – Flawed: Cases include:

  – (Missing Key Points): The answer provides information that only partially meets the question's needs;

  – (Redundant Information): The answer meets the requirements but also contains information not directly useful for the question;

  – The answer is a specific answer to a general question (e.g., question: national policy; answer: local policy);

  – The answer is a general answer to a specific question (e.g., question: local policy; answer: national policy).

• 2 points – Satisfactory: The answer provides information that meets the question's requirements, with no significant irrelevant content.

• 3 points – High Quality: The answer provides information that fully meets the question's requirements, with no redundant information.

**Instructions:**

1. Based on your understanding, first analyze the specific intention and focus of the "Question", as well as any possible extension needs.

2. According to the usefulness definition, provide a comparative analysis of "Answer 1" and "Answer 2".

3. Clearly indicate which answer is overall better (i.e., the "partial order" relationship).

Output the results in the following `JSON` format (strictly follow the requirements, do not output anything extra):

```
{
    "Usefulness Comparative Analysis": "",
    "Final Partial Order": "Answer 1" or "Answer 2"
}
```

Please process the following content according to the above rules:

