# OpenReview forum: "From Faithfulness to Correctness: Generative Reward Models that Think Critically"
_ICLR.cc/2026/Conference — ICLR 2026 Conference Withdrawn Submission_

### Official Review · Reviewer_XPH6 · 2025-10-30

**Soundness:** 2
**Presentation:** 2
**Contribution:** 2
**Rating:** 2
**Confidence:** 4

**Summary:**

The paper proposes a thinking supervised reward model named TRM for open domain QA with retrieved documents. TRM scores each answer sentence in three stages. First faithfulness to sources, then an explicit reasoning step, then a correctness decision. The model is trained with SFT on sentence level supervision and then with RL that uses a combined reward for faithfulness and correctness, with an extra boost for correctly finding incorrect sentences to handle class imbalance. Experiments on a Tencent WeChat search dataset and the CRUD dataset show higher F1 for detecting incorrect sentences and better answer selection than ORM and PRM. When used to train a policy with GRPO and a separate preference model for usefulness, the joint rewards improve correctness and usefulness versus a Qwen2.5 baseline and single reward ablations.

**Strengths:**

- Well defined sentence level supervision that enables finer error localization, with careful annotation protocol that yields the four cases faithful and correct, faithful but wrong, unfaithful but correct, and unfaithful and wrong.
- Comprehensive baseline suite including ORM, PRM, and ablations TRM without reasoning, SFT only TRM, and RL trained TRM. Results consistently favor TRM variants on the main error detection metrics.
- Policy training setup that separates correctness from usefulness by adding a preference model and broadcasting the preference reward to sentences. This is simple and practical, and the joint setup helps both dimensions in different test regimes.

**Weaknesses:**

- The reward model dataset and the policy dataset are both sourced from the same family. The paper says test queries for policy are not seen by the reward model, yet both originate from the same search source which can share artifacts. A stronger split across domains and distributions would help.
- The policy results use GPT 4.1 to label sentence correctness, and preference judgments also rely on GPT 4.1. This can create evaluation bias and circularity. More evaluation on that would be helpful.
- The extra reward for catching incorrect labels and the weight between faithfulness and correctness are central. The paper fixes alpha and the preference weight, and presents one ablation. A sensitivity sweep and calibration analysis would strengthen the claim.
- Since the preference model compares against a single anchor and the judge is a general LLM, the model could win by being longer rather than more helpful. The paper describes order swapping but not length control or toxicity checks.
- Weak experiments. Very few experiments without standard RM benchmarks.

**Questions:**

How robust is TRM to noisy or conflicting documents. Please report per case performance for the four categories faithful and correct, faithful but wrong, unfaithful but correct, unfaithful and wrong on out of distribution data.

How sensitive are results to alpha in the sentence reward and the beta that weights preference reward. Please add a sweep and show Pareto curves for correctness versus usefulness.

Does the explicit reasoning output improve calibrations or error explanations. For example, do faithfulness judgments reduce hallucinated citations or improve edit distance to references.

Can TRM be distilled to a lighter model while retaining most of the gains. Provide a size versus performance plot.

How do you guard against verbosity gaming in the usefulness judge. Did you match token budgets or include length normalization.

Can you compare the proposed method with baselines on standard RM benchmarks?

---

### Official Review · Reviewer_ftTB · 2025-10-30

**Soundness:** 2
**Presentation:** 2
**Contribution:** 2
**Rating:** 4
**Confidence:** 3

**Summary:**

This paper focuses on how to use RLVR to improve the performance of open-domain question answering.
It introduces a Thinking-supervised Reward Model approach, which generates fine-grained sentence-level rewards from the dimensions of faithfulness and correctness.
Experiments demonstrate that such reward signals effectively enhance the identification of incorrect sentences.

**Strengths:**

- This paper focuses on how to improve the performance of open-domain question answering through reinforcement learning, which is a currently interesting and important issue.
- The design of the fine-grained rewards is promising and may be a relatively good direction for improvement

**Weaknesses:**

- The reward for correctness in this paper heavily relies on LLM, which is limited by the knowledge of the LLM used. This may lead to inaccurate rewards.
- The paper distinguishes between faithfulness and correctness but lacks a precise explanation. For example, the case in line 46: "1984 was written by George Orwell in 1949" is incorrect. In my understanding, it is also not faithful, as the document "Novel 1984 was published in 1949" does not directly support this answer. I believe this example does not clearly illustrate the difference between faithfulness and correctness.
- What role does "thinking" play in the design of the reward model? Thinking may introduce significant inference latency. Is it suitable for a reward model? If thinking is not used and a faster model is employed instead, can similar results be achieved?
- The annotated data used in this paper does not seem to be explicitly stated as open-sourced. How can reproducibility be ensured?
- For the QA scenario, why does this paper only focus on cases involving supporting documents? It does not include cases without documents, which seems to be a more common usage scenario for LLMs currently.

**Questions:**

see weakness

---

### Official Review · Reviewer_sCaw · 2025-10-31

**Soundness:** 3
**Presentation:** 2
**Contribution:** 2
**Rating:** 4
**Confidence:** 3

**Summary:**

This paper proposes the Thinking-supervised Reward Model (TRM) to evaluate factual correctness in open-domain question answering.
TRM decomposes judgment into three stages: faithfulness, reasoning, and correctness.
It is trained with supervised fine-tuning and reinforcement learning using faithfulness and correctness rewards.
A two-stage human-annotated dataset separates faithfulness from correctness for each sentence.
Experiments show TRM improves error detection over existing outcome- and process-based reward models.
When combined with a preference model for usefulness, TRM enhances both correctness and usefulness of generated answers.

**Strengths:**

- The paper tackles the important problem of correctness evaluation in open-domain QA, where factual verification is challenging.

- It introduces TRM that explicitly separates faithfulness and correctness.

- Experimental results demonstrate that TRM improves both sentence-level and answer-level error detection.

**Weaknesses:**

- Several sections lack sufficient explanation of experimental settings or rationale behind hyperparameter choices. This may lead to concerns that the method and evaluation were not thoroughly investigated.

- Overall, the paper would benefit from clearer organization and closer alignment between figures, tables, and textual analysis:

1. The paper does not explain why alpha in Eq (2) is fixed at 0.5, nor does it include an ablation or sensitivity analysis to justify this choice.

2. The data construction process lacks key details such as the total number of annotators, annotations per sentence, and inter-annotator agreement.


3. Line 407-408: Correctness and usefulness evaluations rely solely on GPT-4.1 as the judge, without testing multiple evaluators or cross-model agreement.


4. Line 416: Usefulness evaluation uses Qwen2.5-32B-Instruct as the only anchor, which may bias results. No alternative anchors or robustness tests are reported.


5. Line 674-675: The 1:2 weighting between TRM and the preference reward model during policy training is not explained or experimentally validated.


6. Line 315-316: Table 2a and 2b are referenced unclearly. The text does not specify which subtable corresponds to which result.


7. Table 1 is placed far from the main discussion, making it difficult to connect the table with the analysis.


8. Figure 3 appears in the paper without any in-text reference or discussion, which reduces interpretability.

**Questions:**

- Why is alpha fixed at 0.5 in Eq (2)?

- Lines 407–408: Why is GPT-4.1 the only evaluator? Would it be better to include multiple models for consensus?

- Line 416: Why is Qwen2.5 used as the sole anchor for usefulness evaluation? Would using multiple anchors increase reliability?

- Lines 674–675: Why is the reward weighting ratio set to 1:2?

- Lines 315–316: What does Table 2a refer to? Which part of Table 2 corresponds to (a) and which to (b)?

---

### Official Review · Reviewer_ZTiy · 2025-10-31

**Soundness:** 2
**Presentation:** 2
**Contribution:** 2
**Rating:** 2
**Confidence:** 3

**Summary:**

RL with Verifiable Rewards (RLVR) works well in domains like math and coding, where correctness is easily verified. In open-domain QA, verifying correctness is hard due to ambiguous or conflicting information. Existing work has primarily focused on improving faithfulness, defined as semantic alignment with supporting documents, but not focused on improving factual accuracy, leading to errors.

To address this, authors proposed Thinking-supervised Reward Model (TRM). TRM evaluates answers in three steps: Given a query, answer, and supporting documents, TRM first evaluates the sentence-level faithfulness of the answer to the provided evidence. In the 2nd step, TRM will assess how does faithfulness inform correctness. Finally, TRM will check if the sentence is factually accurate. Results show that TRM outperforms baseline reward models (ORM and PRM) in detecting incorrect sentences and answers. Incorporating TRM into reinforcement learning improves correctness by up to 30.3% and usefulness by up to 35%. Ablation studies confirm the importance of the reasoning step.

**Strengths:**

The improvements on correctness and usefulness seems substantial.

The paper is clearly written and easy to follow

**Weaknesses:**

The evaluation is only on Tencent and CRUD datasets. Tencent dataset is closed source and not publicly available. The paper does not demonstrate generalizability across diverse domains, such as [HotpotQA](https://arxiv.org/abs/1809.09600), [AmbigQA](https://arxiv.org/abs/2004.10645).

The paper reports aggregate metrics but lacks qualitative analysis of where TRM fails (e.g., misjudging correctness despite high faithfulness). Please include error analysis and categorize common failure patterns.

How accurate is the judge? Usefulness is evaluated using LLM-as-a-judge (GPT 4.1), not human annotators. Please add human judges

While TRM introduces structured reasoning, similar ideas of decomposing correctness and faithfulness exist in prior works on fact-checking and verifiable QA.
Adlakha et al., 2024
Evaluating Correctness and Faithfulness of Instruction-Following Models for Question Answering
Metropolitansky et al., 2025
Towards Effective Extraction and Evaluation of Factual Claims

**Questions:**

The example is line 046 seems like an issue for retriever. It retrieves an irrelevant text “Novel 1984 was published in 1949”  as the context. But the author claimed that this is the error of the verification.

---

### Note · Authors · 2025-11-20

I have read and agree with the venue's withdrawal policy on behalf of myself and my co-authors.